# Modulation of Cellular Response to Different Parameters of the Rotating Magnetic Field (RMF)—An In Vitro Wound Healing Study

**DOI:** 10.3390/ijms22115785

**Published:** 2021-05-28

**Authors:** Magdalena Jedrzejczak-Silicka, Marian Kordas, Maciej Konopacki, Rafał Rakoczy

**Affiliations:** 1Laboratory of Cytogenetics, West Pomeranian University of Technology in Szczecin, Klemensa Janickiego 29, 71-270 Szczecin, Poland; mjedrzejczak@zut.edu.pl; 2Faculty of Chemical Technology and Engineering, West Pomeranian University of Technology in Szczecin, Piastow Avenue 42, 71-065 Szczecin, Poland; mkordas@zut.edu.pl (M.K.); mkonopacki@zut.edu.pl (M.K.)

**Keywords:** rotating magnetic field, magnetic induction, frequency, keratinocites, fibroblasts, cellular metabolic activity, ROS level, Ca^2+^concentration level

## Abstract

Since the effect of MFs (magnetic fields) on various biological systems has been studied, different results have been obtained from an insignificant effect of weak MFs on the disruption of the circadian clock system. On the other hand, magnetic fields, electromagnetic fields, or electric fields are used in medicine. The presented study was conducted to determine whether a low-frequency RMF (rotating magnetic field) with different field parameters could evoke the cellular response in vitro and is possible to modulate the cellular response. The cellular metabolic activity, ROS and Ca^2+^ concentration levels, wound healing assay, and gene expression analyses were conducted to evaluate the effect of RMF. It was shown that different values of magnetic induction (*B*) and frequency (*f*) of RMF evoke a different response of cells, e.g., increase in the general metabolic activity may be associated with the increasing of ROS levels. The lower intracellular Ca^2+^ concentration (for 50 Hz) evoked the inability of cells to wound closure. It can be stated that the subtle balance in the ROS level is crucial in the wound for the effective healing process, and it is possible to modulate the cellular response to the RMF in the context of an in vitro wound healing.

## 1. Introduction

Electromagnetobiology or electrobiomagnetism is a branch of science whose main goal is to analyze the influence of electromagnetic field (EMF) on living organisms. The electric and magnetic fields are components of EMF. The electric field (EF) is generated by the presence of an electric charge, and this kind of field defines the magnitude and direction of the force it exerts on a positive electric charge. The potential between charge-carrying bodies determines the magnitude of EF. The magnetic field (MF) is created by means of the motion of electric charges, and this kind of field acts only on electric charges in motion. The magnitude of MF is proportional to the current flow in a conductor. The EMF can be organized taking into consideration their frequency or wavelength. Generally, the EMF may be divided into extremely low (0–300 Hz), low, middle, and high (30 kHz–30 MHz), ultra-high (30–300 MHz), and super high (300 MHz–30 GHz) [1,2,3]. Hunt et al. (2009) showed that electromagnetic stimulation takes into account the influence of predominantly magnetic fields, predominantly electric fields, or fields with both electric and magnetic components [4].

In recent years, there has been an increasing interest in the stimulation of living cultures using the various types of MFs. One major issue in the application of MF for biotechnology concerned the method of creating a magnetic field and its correct quantitative evaluation as a factor affecting living organisms, their tissues, and cells. The static or stationary MF (constant MF that do not change in time) or alternating MF (MF varying with time) may be applied to stimulate the microorganism. The static magnetic fields (SMFs) can be divided into weak (<1 mT), moderate (1 mT to 1 T), strong (1–5 T), and ultrastrong (>5 T) [4,5,6]. Practically, the MFs changing with time or form one or several spatial coordinates may be classified due to different operation modes, such as steady-state (no time-varying), slowly varying (extremely-low frequency range) in time, and pulsing mode [1,2,3]. Table 1 illustrates the systemization of MFs taking into consideration their changes with time and space.

It can be assumed that the MF as a manifestation of the EMF is the space surrounding a magnet or a material in which the current flows. It is caused by moving electrical charges and is characterized by the moving through the charged particles on which the force is acting. The MFs can be divided into two main types: direct current (DC) MF (DCMF) and alternating current (AC) magnetic field (ACMF). The main feature of DCMF is that it does not change with time or changes very slowly. Such fields do not have the frequency (MF vector is constant in time and space). The static magnetic field (SMF) is an example of MF caused by the constant current. It should be noted that the ACMF varies with the frequency. A pulsating magnetic field (PMF) may be treated as a typical example of ACMF. The PMF has an external MF vector which changes as a sine-wave. This vector changes at every point of the space and with time. Another example of the ACMF is a rotating magnetic field (RMF). The RMF may be generated by the superposition of three 120° out of phase PMFs. This field has a constant intensity over time while it changes its direction continuously at any point of the domain (RMF is variable in space). The RMF is commonly used in AC motors and is generated by a stator motor [4,6].

Since the effect of MFs on various biological systems has been studied, different results (even contradictory) have been obtained and presented in many valuable works, e.g., the weak MFs (<1 mT) were specified as insignificant to evoke biophysical mechanisms that interact with biological samples [5,7]. In some studies, it was found that the magnetic responses (expressed in the enzymatic reactions catalyzed by enzymes such as ATP synthase, phosphoglycerate kinase, pyruvate kinase, and creatine kinase) are only observed under nonphysiological conditions except for magnetoreceptors [7,8,9]. On the other hand, some studies present that the WMFs (weak magnetic field) affect the biological system; for example, Lacy-Hulbert et al. (1995) presented that WMF exposure affected human leukemia cells that demonstrated higher levels of the transcription factor c-Myc (major effect of c-Myc is B cell proliferation and has been associated with B cell malignancies) [10,11,12]. Another study [13] conducted by Vanderstraeten et al. (2012) suggests that the ELF magnetic fields may disrupt circadian timing in the circadian clock system, which results in changes in the expression of circadian genes, which are linked with regulatory genes of the cell cycle (e.g., cellular responses to DNA damage, such as repair, cell-cycle checkpoints, and apoptosis) [14,15,16] and various cancer-relevant cellular processes [13,17,18]. Thus, the hypothesis of magnetosensitivity of the cellular circadian system seems to be linked with relevance to cancer [18].

In contrast, in another study [19] Novoselova et al. (2019) demonstrated a strong suppression of tumour (solid tumors induced by Ehrlich ascitecarcinoma (EAC) cells induced in mice) growth (mouse exhibited higher survival rate and longer average lifespan) evoked by the weak combined constant magnetic field (60 μT) and an alternating magnetic field (100 nT) containing six frequencies (from 5 to 7 Hz). The suppression of tumour growth after exposure of the mice to moderate magnetic fields (100 mT, 1-Hz, 360 min daily for as long as 4 weeks) [20] was also observed by Tatarov et al. (2011). Whereas the effect of weak magnetic field on biological systems is still under debate, the stronger magnetic fields exhibit a pronounced effect on biological systems. For example, static magnetic fields (SMFs) were described as freely penetrating biological tissues and interacting directly with moving charges, such as proteins, ions, and magnetic materials present in tissues through several physical mechanisms. SMF induces, among others, proinflammatory changes, an increase of reactive oxygen species (ROS), free radicals (via Fenton reaction) generation, and radical pair recombination [21,22,23]. It is noteworthy that magnetic fields, electromagnetic fields, or electric fields may be used in medicine. Tatarov et al. (2011) suggested that a moderate magnetic field demonstrates potential in cancer therapeutics as an adjunct or primary therapy. On the other hand, MFs and/or EMFs are used in wound-healing stimulation, soft tissues and bones regeneration, treating disorders of the central nervous system, and reducing pain [24,25,26]. 

In many cases, the magnetic field is considered a therapeutic agent that promotes the wound healing process. A lot of studies try to present a hypothesis about the mechanisms of the influence of the magnetic field on the wound healing processes. The main assumptions are as follows. (i) WMF can evoke new tissue production/regeneration (stem cell proliferation and subsequent differentiation) due to manipulation of ROS levels and also downstream heat shock protein 70 (Hsp70) expression [5]. (ii) The magnetic field causes changes in membrane potential and temporary membrane permeabilization that affects sodium content and potassium-efflux or the transmembrane voltage [1,2,27]. (iii) The calcium gradient between the extracellular and intracellular fluid is a transduction second messenger [28], and its gradient could potentially be affected by EMFs and MFs. Moreover, the RMF exposition may be responsible for the activation of the store-operated calcium entry (SOCE) pathway and Ca^2+^ distribution to pass across the plasma membrane [1,2,3,29]. (iv) MF may induce changes in enzymatic activities (e.g., enzymes involved in mitochondrial metabolism). (v) MF may cause cytoskeletal organization (due to reorganization of the electrostatically negative charged actin filaments), and those changes may affect the cellular shape, endoplasmic reticulum, mitotic apparatus [2,3]. (vi) Finally, the RMF creates the mixing process at the micro-level and may affect the energy level; some of the selected molecules strongly influence the transfer processes between the living cells and the culture medium [1,2,3,29].

In the presented study, we sought to determine whether low-frequency RMF with different field parameters (different magnetic induction and frequency values) could evoke the cellular response in vitro (L929 and HaCaT cell culture models), and is it possible to modulate the cellular response to different parameters of the RMF in the context of an in vitro wound healing.

## 2. Results

### 2.1. Relative Cell Metabolic Activity/Viability Based on CCK-8 and NRU Assays

To determine whether low-frequency RMF (with different parameters such as magnetic induction and frequency values) evoke the cellular response of L929 and HaCaT cell culture in vitro, the metabolic activity of cells, as an indicator of cell viability, was analyzed using the Cell Counting Kit-8 (CCK-8). The conducted analyses in the presented study highlight a cell-specific response to different parameters—magnetic induction (*B*) and frequencies (*f*) values of RMF. It is well known that the CCK-8 assay can provide two pieces of information—the determination of the metabolic activity of cells (cell viability) and proliferation of cells. Both information is based on bioreduction of WST-8 (Water Soluble Tetrazolium 8) salt to an orange formazan due to cellular dehydrogenase activity and NADP(H) and NAD(H) levels. 

In our study, the metabolic activity of cells estimated by the CCK-8 assay exhibited different tendencies due to cell type exposed to the magnetic flux density (Figure 1a). The cultures of L929 cells demonstrated higher metabolic activity for *f* = 30 Hz and *B* = 13.6 and 28.4 mT (Figure 1a), whereas for *f* = 50 Hz cells, the metabolic activity values were similar for each tested magnetic induction *B* (with significant differences, *p*-values < 0.05. In the case of HaCaT cell cultures (Figure 1b), the highest metabolic activity (≈40% higher than control samples) was observed for *f* = 30 Hz and all applied magnetic induction values. In contrast, the HaCaT cells exposed to RMF with a frequency of 50 Hz (Figure 1b) showed the lowest metabolic activity in comparison to cells exposed to 30 Hz, with the lowest value for 22.8 mT (with significant differences, *p*-values < 0.05. The higher values that in control cultures both for L929 and HaCaT cells obtained from CCK-8 might be generated by a higher number of cells (due to the proliferation process) or by the potential increase in enzymatic dehydrogenase activity. When we take into account that the proliferation process takes 21–24 h in the case of L929 cells and 28 h in the case of HaCaT cells, in our study, we were not able to observe changes in the proliferation process due to a short time from RMF exposition (4 h of exposition) to result obtained from CCK-8 assay analysis (3 h of assay incubation). That is why we suggest that the higher values indicated in CCK-8 may be related to the potential increase in enzymatic activity, although the method used is generally considered not as specific for enzymatic activity.

To investigate the cellular response to tested RMF, additionally, neutral red uptake assay was performed. The minimal changes were found in the ability of L929 (Figure 2a) and HaCaT (Figure 2b) cells to incorporate neutral red in lysosomes (significant differences, *p*-values < 0.05.

### 2.2. Reactive Oxygen Species (ROS) Generated by RMF

Our study demonstrates that the cellular response in the reactive oxygen species (ROS) level was different between L929 and HaCaT cell cultures. The L929 fibroblasts responded to the applied rotating magnetic field (*f* = 30 Hz; *B* from 5.9 to 28.4 mT) with lower ROS accumulation in comparison to control cultures. A frequency of 50 Hz evoked an increase of ROS levels with the highest results for magnetic induction (*B* = 5.9, 18.6 and 28.4 mT) (Figure 3a). For the HaCaT cell line (Figure 3b), we noticed two-fold higher ROS levels relative to control (significant differences, *p*-values < 0.05). The highest ROS generation was observed for RMF parameters such as *f* = 30 Hz and two values of magnetic induction *B* = 5.9 and 13.6 mT. The RMF with *f* = 50 Hz and *B* = 5.9 and 16.6 mT also affect ROS accumulation. Moreover, on the example of HaCaT cells, it was noticed the tendency where the ROS levels are inversely proportional to the magnetic induction values.

### 2.3. Changes of Intracellular Concentration of Ca^2+^evoked by RMF

In the presented study, we also determined the effect of RMF (with different parameters) on changes of the intracellular concentration of Ca^2+^ in L929 and HaCaT cultures (Figure 4a,b). The L929 cell culture observations indicated that RMF with frequency *f* = 30 Hz evoked relatively small changes (approximately 10% higher relative to control) in intracellular concentrations of Ca^2+^. Whereas, L929 cell cultures incubated in RMF (*f* = 50 Hz) displayed a decrease of Ca^2+^ level with the increasing magnetic induction. Results obtained from HaCaT cell cultures showed a higher intracellular concentration of Ca^2+^ (the highest for B = 18.6 mT and *f* = 30 Hz) and relatively small changes in Ca^2+^ ions concentrations for *f* = 50 Hz relative to control samples (significant differences, *p*-values < 0.05. Comparison of intracellular Ca^2+^ ion concentration between L929 and HaCaT cell cultures highlighted the cell-specific response and displayed different changes in cellular Ca^2+^ ions metabolism evoked by different RMF parameters (Figure 4a). 

Additionally, changes in ROS levels may be correlated with changes in intracellular Ca^2+^ ion levels especially in the case of HaCaT cell cultures. It was found that HaCaT cells exposed to RMF (B = 28.4 and both *f* values) exhibited the lower ROS level and simultaneously the lower cellular Ca^2+^ ions level.

### 2.4. Effect of RMF on Wound Healing Process

The intracellular concentration of Ca^2+^ and ROS levels may affect the cell migration; thus, the cell migration process into the wound was observed in our study (Figure 5).

When the L929 cells were exposed to RMF (*f* = 30 Hz), a minimally lower migration rate was observed for magnetic induction *B* = 13.6 mT, whereas for other magnetic induction values, the wound closure process was slightly higher relative to control samples after 24 h from the exposure (Figure 6a). When L929 cells were exposed to RMF with parameters 5.9–28.4 mT/50 Hz, the cell migration ability and migration rate were minimally higher (in comparison with control, except for 28.4 mT sample) in a dose-dependent manner (significant differences, *p*-values < 0.05). The lower magnetic induction had the highest rate in the wound healing process (Figure 6b). In the case of HaCaT cells, the RMF (*f* = 50 Hz; *B* = 13.6 and 28.4 mT) affected cell migration ability, which was slower in contrast to L929 cells. The wound closure process efficiency was higher (relative to control samples) in the human keratinocytes cultures (HaCaT cell line) at 22.8 mT of magnetic induction for frequency 50 Hz (Figure 6c,d).

Moreover, we noticed some correlation between intracellular calcium ion levels (Figure 4a) and wound healing efficiency (Figure 6c) in L929 cells. The lowest intracellular Ca^2+^ ion concentration (in RMF *f* = 50 Hz) had the lowest cell migration ability in the wound closure process. The human keratinocytes exposed to RMF (*f* = 50 Hz and *B* = 13.6 and 28.4 mT) exhibited the lowest ability in wound healing, and those cultures displayed a lower ROS level (but highest that in control samples) and lower intracellular Ca^2+^ ion concentration. To validate our observation about the correlation between intracellular calcium ion levels, ROS levels, and wound healing efficiency, the multiple correlation coefficient (*R*) was applied. The relationship between the WHA value for the various time points (0 h—measurement directly after exposure to RMF; 24 h—measurement 24 h after exposure to RMF; 48 h—measurement 48 h after exposure to RMF) and a linear combination of ROS and Ca^2+^ values were presented in Figure 7. This coefficient yields the maximum degree of linear relationship that can be obtained between two independent variables (ROS and Ca^2+^) and a single dependent variable (WHA at 0 h; WHA at 24 h; WHA at 28 h). In the case of the L929 cell culture after 24-h exposition to RMF (at magnetic inductions equal to 5.91, 13.60, and 28.41 mT and frequency equal to 30 Hz), the correlation between the ROS, Ca^2+^ levels, and wound healing closure efficiency was found. In contrast, the RMF at a frequency equal to 50 Hz evoked changes in wound healing closure efficiency that correlate with ROS and intracellular Ca^2+^ levels at magnetic induction equal to 18.6, 22.79, and 28.41 mT. The second cell line—HaCaT cells—directly and 24 h after the RMF (at frequency 30 Hz) incubation, exhibited changes in ROS and intracellular Ca^2+^ levels, and those changes were correlated with wound healing closure efficiency and magnetic induction equal to 5.91, 18.64, 22.79, and 28.41 mT (Figure 7).

Finally, in the case of the RMF at a frequency equal to 50 Hz, correlation between changes in the ROS and intracellular Ca^2+^ levels and wound healing closure efficiency was found for all analyzed time points but only for magnetic induction equal to 5.91 and 28.41 mT.

### 2.5. The RMF and Its Effect on Gene Expression Levels

In our study, we also analyzed the effect of different parameters of RMF on genes that were chosen for the presented study. In general, selected genes can be divided into two groups due to their role in the wound healing process. Genes from the first group (*cola1a1, cola3a1, KRT10, KRT14*) are involved in, among others, proteins synthesis, ECM deposition, and keratinocyte proliferation [30,31,32]. In the second group were genes involved in cellular migration and wound closure (cdc42, *Rac1*) [33,34].

In normal skin, two types of collagen (I and III) coexist in a ratio of approximately 4:1 [35]; thus, in our study, the procollagen type I (*col1a1*) and the procollagen type III (*col3a1*) genes were analyzed in L929 cells exposed to RMF. For the procollagen type I gene, the average Ct values were in the range of 15.97 to 17.74, whereas Ct for the procollagen type III gene was in the range of 19.57 to 21.54. The average relative *col3a1* gene expression (the average relative quantity—RQ; normalized using *GAPDH* values) was lower after exposition to RMF for all analyzed parameters (in relative to control samples). The highest *col1a1* gene expression was observed in cell cultures exposed to RMF with parameters 13.6 mT/50 Hz (Figure 8) (higher than in control samples), but it was significantly lower in all other experimental groups for *f* = 50 Hz (Figure 8) (significant differences, *p*-values < 0.05.

The HaCaT cell cultures were analyzed for the average relative expression of keratin 10 (*KRT10*) and 14 (*KRT14*) genes. The average relative expression of *KRT10* and *KRT14* genes were lowest in all experimental groups (except 22.8 mT/30 Hz) relative to control samples (significant differences, *p*-values < 0.05). The relative expression of both genes presented a sinusoidal waveform trend, with the highest *KRT10* and *KRT14* gene levels for HaCaT cells incubated for 4 h in RMF with parameters 22.8 mT/30 Hz (Figure 8). 

Moreover, the relative expression level of *cdc42*/*CDC42* and *rac1/RAC1* genes in both cell lines were analyzed. The highest RQ was found for *cdc42* in L929 cells exposed to RMF (*B* = 5.9 mT and *f* = 30 Hz) and *CDC42* in HaCaT cells (*B* = 5.9 mT and *f* = 30 Hz) (Figure 8). In HaCaT cells, the relative expression level of *CDC42* gene is inversely correlated with ROS level (significant differences, *p*-values < 0.05. 

Another factor important in the wound healing process is Rac1. The mentioned factor is a protein called an effector of cellular responses to growth factors, cytokines, and adhesion proteins present in wounds. Fibroblasts–L929 cells exhibited the highest RQ for magnetic induction *B* = 13.6 mT (*f* = 30 Hz), whereas HaCaT cells displayed the highest for magnetic induction *B* = 5.6, 13.6, and 22.8 mT (*f* = 30 Hz). The culture of both cell lines exhibited lower RQ in all experimental groups relative to control samples (significant differences, *p*-values < 0.05 for RMF with *f* = 50 Hz (Figure 8)).

## 3. Discussion

### 3.1. Rotating Magnetic Field Affect Cell Viability in Different Manner

The cellular response of L929 and HaCaT cell culture in vitro to different parameters—magnetic induction (*B*) and frequency (*f*) values of RMF was analyzed by determining the metabolic activity, and our findings highlight a cell-specific response. Although both L929 and HaCaT cultures demonstrated higher values of the metabolic activity for *f* = 30 Hz, the HaCaT demonstrated much higher metabolic activity than the L929 cells and control samples. Thus, we may state that HaCaT cells are more sensitive to applied RMF parameters than L929 fibroblasts. The proliferation process underlines that the higher cellular response to RMF possible was excluded due to a short time from RMF exposition to results obtained from CCK-8 assay analysis; thus, we suggest that the observed tendency is potentially related to the increased enzymatic dehydrogenase activity (in CCK-8 assay) after RMF exposition. It should be taken into account that generally, CCK-8 is not a specific method to determine dehydrogenase enzymatic activity, but in 2019, Chamchoy et al. presented that the WST-8 salt (used in CCK-8 assay) is suitable for the measurement of NAD(P)H and dehydrogenase activity. The obtained results were verified by the use of glucose-6-phosphate dehydrogenase (G6PD) and glucose dehydrogenase (GDH) to measure enzymes activity. It was found that WST-8 is sensitive and could serve as a high-throughput method for measuring dehydrogenase activity [35]. We have to take into account that the sensitivity of CCK-8 involves the dehydrogenases (but also NAD(H), NADP(H), and mitochondrial activity) in cells; thus, it might be stated that changes in cells after RMF exposition are related to higher general cell enzymatic (metabolic) activity. 

Moreover, our findings were in agreement with the results obtained by Rusak and Rybak (2013). In the mentioned study, the AC magnetic field evoked the increased mitochondrial reductase activity in conversion of MTT to formazan compared to control in Balb 3T3 cell cultures [25]. The same findings were presented by Yamashita and Saito (2001) and indicated that the mitochondrial energy activity was enhanced and the cell respiration was increased after exposure to the moderate static magnetic field (100 mT) [36]. It was also found that WMFs produce transient induction of the membrane permeability transition [37] and thus increase cytosolic cytochrome c levels in human amniotic cells (via an increase in ROS) [5,37].

In the case of other studies, static magnetic field—SMF (250 mT, during 3 h) altered the intracellular labile zinc fraction in THP1 cells; thus, in consequence, SMF might indirectly alert superoxide dismutase (SOD) activity (higher SOD activity relative to control) [38]. Whereas, another study based on the cancerous and noncancerous human gastric tissues demonstrated that the static magnetic field evoked an increase in SOD activity and decrease in malondialdehyde (MDA, a marker for oxidative stress) level in the noncancerous tissue, but decreases of SOD and glutathione peroxidase (GSH-Px) activities and increases MDA level and catalase (CAT) activity in the cancerous tissue [39]. Moreover, Azadniv et al. (1995) and Mullis et al. (1999) find that cellular response to the magnetic field was manifested by enhanced ornithine decarboxylase activity [40,41]. It was also found that low frequency pulsed magnetic fields modulate the activity of the protein kinases and ornithine decarboxylase with the result in an increase of their activity [42].

Moreover, the cellular response (cell viability) to tested RMF was examined by neutral red uptake assay, and it was found that the ability of L929 and HaCaT cells to incorporate neutral red in lysosomes (permeabilization of lysosomal compartments) was not affected by RMF. The same effect was observed in another study [43] by Romeo et al. (2016), where the absence of effects on MRC-5 human fetal lung fibroblasts viability exposed to 370 mT of the SMF. Briefly, it can be stated that the applied RMF evoked changes in the enzymatic activity of L929 and HaCaT cells, whereas not alert lysosome function.

### 3.2. ROS and Intracellular Ca^2+^ Changes Generated by Rotating Magnetic Field—Cell Type Matters

Many studies demonstrate that WMFs can affect biological systems in many different ways [5] and give evidence of the significant impact of different magnetic fields on the reactive oxygen species (ROS) level [44]. The reactive oxygen species (ROS) were found to be generated by MF, but the role of ROS in cellular processes is complex and unclear. Most of the studies found that SMFs increases ROS levels, while other studies give evidence that the ROS level remains unchanged [45]. When the human neuroblastoma SH-SY5Y cell cultures were exposed to the static magnetic field (SMF), it was found that different intensities evoke the intercellular ROS increase [46,47]. Other examples presented the effect of SMF to increase endogenous ROS in embryoid bodies and cardiac progenitor cells derived from mouse ES cells [48,49]. Moreover, ROS was indicated to play a role in the vasculogenesis and cardiomyogenesis of mouse embryonic stem (ES) cells [50,51]. There is some other evidence that the reactive oxygen species (ROS) has a beneficial impact on the wound healing process. For example, increasing ROS may lead to inhibition of pathogen colonization, regulation of angiogenesis, and promotion of effective tissue repair. The role of ROS in regeneration and new tissue formation in vivo was observed on the planarian regeneration model by Huizen et al. (2019). Interestingly, stem cell proliferation and subsequent differentiation (in vivo planaria model) can be manipulated via reactive oxygen species (ROS) accumulation and downstream heat shock protein 70 (Hsp70) expression [5]. It was also found that the strength of WMF played a crucial role in increasing or decreasing new tissue formation via the modulatory effect of ROS levels; thus, WMF may be a potential therapeutic tool [5]. The idea to use MFs as therapeutic tools is supported by the results of the investigation based on noncancerous fibroblast (control) and fibrosarcoma cell cultures. It was demonstrated that WMF caused reduced growth of fibrosarcoma cell (highly proliferative cells) but did not affect the growth of noncancerous fibroblasts [51]. The reduction of human leukemic cells (K562) proliferation was obtained by combined paclitaxel treatment and moderate SMFs (*B* = 8.8 mT) exposition [52]. In addition, Chen et al. (2010) found that the static magnetic fields enhanced the potency of cisplatin on K562 cells, resulting in DNA damage [53]. The same findings were observed in vivo by others, where tumours’ growth suppression in mice was demonstrated [19,20].

The excessive studies focused on showed that ROS is a part of normal cell physiology, including intracellular signal transduction and the role of ROS in cell signaling [18,54], but on the other hand, the excessive ROS release can damage healthy cells and impair the wound repair process [55,56]. Okano (2008) stated that the SMF increases the concentration of free radicals (that escape from the radical pair) that affect membrane damage and cell lysis [57]. It was also presented that ELF magnetic fields may affect the circadian clock system and evoke changes in regulatory genes of the cell cycle [14,15,16]. Some other observations indicate that ELF magnetic fields alter DNA damage responses as well as affect ROS-related cellular processes and the induction of genomic instability in cells (exposed to 50–60 Hz magnetic fields at 100–1000 mT [18].

The intracellular Ca^2+^ is an important second messenger that regulates various cellular functions, including secretion, contraction, metabolism, gene expression, cell survival, and cell death [58,59]. The intracellular Ca^2+^ ion concentration and its changes influence cellular morphology, the polymerization of microtubules, and calcium-dependent assembly of actin filaments; thus, they indirectly may affect the wound healing process [60,61]. The Ca^2+^ ions homeostasis in the mitochondria plays an important role in cellular physiology and pathophysiology. Calcium communicates with reactive oxygen species (ROS), hydrogen peroxide (H_2_O_2_), or hydroxyl radicals (HO^•^) [60]. In the plasma membrane, several calcium transporters are localized that can be regulated by ROS; moreover, ROS and calcium are mutually interconnected. Calcium can increase the production of ROS, and ROS can significantly affect calcium influx into the cell [60].

### 3.3. Wound Healing Assay—Migration of Cells and Gene Expression Analyses

Changes in the intracellular concentration of Ca^2+^ and ROS level evoked by MF may affect the cell migration process; thus, the cell migration process into the wound was observed in our study. The RMF affected the wound closure process slightly, and the L929 cells exhibited a higher migration rate relative to control samples 24 h after exposure (Figure 7a). A different response was observed for the human keratinocytes’ exposed to RMF, which exhibited the lowest ability to migrate in wound healing. This finding might be explained by ROS and Ca^2+^ ion concentration. The lower migration ability of HaCaT cells might be co-regulated by lower ROS and intracellular Ca^2+^ ion concentration levels.

In this study, the expression of genes that play an important role in the cell migrations and the extracellular matrix (ECM) synthesis was determined. Acute wounds healing is a complex process and defined by phases such as coagulation (hemostasis), inflammation (with phagocytosis), matrix protein synthesis and deposition, angiogenesis, fibroplasia, epithelialization, wound contraction, and tissue remodeling (maturation) [60]. It is known that fibroblasts appear in the wound within 2 to 3 days, play a crucial role in the wound closure, and those cells produce substances important to the wound repair process, including glycosaminoglycans (GAG) and collagen. The principal role of collagen is to act as a scaffold in connective tissue (type I, II, and III forms), and the wound during the fibroblastic phase is increasing wound tensile strength [30,61]. Haukipuro et al. (1991) demonstrated the synthesis of both type I and type III procollagen during human wound healing, thus stimulating the effect of RMF on *col1a1* and *col3a1* genes, which under consideration. In performed experiments, only in two cases, was a higher expression level noticed. The RMF at magnetic inductions equal to 13.60 mT and frequency equal to 30 Hz induced a higher expression level of the *col3a1* gene, whereas at magnetic inductions equal to 13.60 and frequency equal to 50 Hz, a higher expression level was found in case of the *col1a1* gene. Unfortunately, none of RMF’s parameters evoked the desired changes in both *col1a1* and *col3a1* genes in L929 cell cultures.

In the wound healing process, keratinocytes are involved, among others, in wound edges formation. The KRT10 is expressed in post-mitotic, differentiating, suprabasal keratinocytes within the regenerating epidermis differentiated keratinocytes. It was also found that in the wound, proximal keratinocites (KCs) downregulate KRT10-K1 upon skin injury. On the other hand, the KRT14 is expressed by mitotically active, basal keratinocytes, and in the wound, proximal KCs upregulate KRT16/KRT17-KRT6 [32,60,61]. In our study, only the RMF frequency equal to 30 Hz induced the highest expression of *KTR10* and *KRT14* for 22.8 mT magnetic induction. In the perfect situation, we should obtain lower expression for the *KTR10* gene and higher for the *KTR14* gene, but we did not obtain appropriate results in HaCaT cells.

The GTP-binding proteins such as Rho, Cdc42, and Rac are known as the Rho family and play a crucial role in regulators of the cytoskeleton [62]. The protein Cdc42 is an important regulator of wound repair due to its role in the cell division cycle. Moreover, Cdc42 may interact with receptor tyrosine kinase (RTK) that participates in cell migration and cell–cycle progression. Furthermore, experiments with silencing of *Cdc42* expression in cultures caused inhibition of wound closure due to a decrease in epithelial migration and growth. Cdc42 is an important regulator of corneal epithelial wound repair [62]. In our study, positive changes in *CDC42/cdc42* gene expression (higher expression level) were confirmed only in the case of RMF with parameters 5.9 mT of magnetic induction and 30 Hz of frequency. 

*Rac1* has long been suspected to be an important regulator of wound healing. Additionally, the essential role of Rac1/RAC1 in wound healing is promoting keratinocyte migration and proliferation during wound re-epithelialization [63]. Desai et al. (2008) [64] showed that the coordination of both RhoA and Rac1 activity contributes to bronchial epithelial wound repair mechanisms in vitro [64]. The mentioned coordination is based on the inhibition of Rho-kinase that accelerates wound closure and the activation of Rac1 required for normal epidermal wound healing in two ways: the regulation of keratinocyte proliferation and migration [65]. In analyzed samples, the relative expression was higher than in control in the case of L929 exposed to RMF with 13.6 mT of magnetic induction (*f* = 30 Hz). Interestingly, all magnetic induction values of the RMF (*f* = 30 Hz) induced higher relative expression *RAC1* gene in HaCaT cell culture. Moreover, our results with higher expression of the *RAC1* gene correspond to a better WHA process of HaCaT cells in case of frequency 30 Hz of RMF.

### 3.4. Hypothesis of Mechanism of the Influence of the Magnetic Field on the Wound Healing Process

Based on all experiments and obtained results, we selected the RMF at frequency equal to 30 Hz that evoked better changes (e.g., higher metabolic activity of HaCaT cells for all magnetic induction values; more effective wound healing process in L929 and HaCaT cells) in both L929 and HaCaT cultures than the RMF at a frequency equal to 50 Hz. When L929 fibroblasts were exposed to RMF at a frequency equal to 30 Hz, cells did not exhibit metabolic activity as high as HaCaT cells. The stimulatory effect after exposure (for 4 h and less) to the magnetic field was demonstrated by CCK-8 assay based on Schwann cells (SCs). Liu et al. (2015) provided evidence that PMF influence SCs’ proliferation, viability, and biological properties. In addition, PMF enhanced neurotrophic factors secretion [66]. A higher cellular metabolic activity might be connected with two-fold higher ROS levels relative to control in HaCaT cells, whereas L929 cell cultures exhibited lower ROS accumulation in comparison to control cultures. These data suggest that there is higher cell metabolic activity at the highest the ROS level. Furthermore, ROS generation (as was demonstrated by the multiple correlation coefficient, R) is related to intracellular calcium ion levels and wound healing efficiency (in RMF *f* = 30 Hz). It is known that reactive oxygen species (ROS) are generated in many cellular components (by multiple cellular processes) as by-products by normal cells and play an important role in signalling pathways. ROS production is controlled by changes in metabolic and signalling pathways, and normal cells maintain oxidative homeostasis. The ROS generated by RMF should be at levels that do not affect the mentioned oxidative homeostasis; a permanent increase in ROS levels can promote cell death and develop pathological states. That is why the therapeutic effect might be achieved over a range and without affecting the subtle balance of reactive oxygen species levels [67]. Moreover, there is a general conception that Ca^2+^ overload leads to stimulated ROS generation in mitochondria, but the literature demonstrated also the opposite thesis [68]. Other studies also confirmed the important role of intracellular (Ca^2+^) in physiological processes; e.g. in the research of Yap et al. (2019), brief exposure of C2C12 myoblasts to pulsed magnetic fields evoked an augmented increase of cytosolic calcium intracellular (Ca^2+^) concentration; moreover, simultaneously, pulsed electromagnetic field (PEMF) stimulated the production of ROS and ATP [69]. There were positive effect of PEMFs on increments in myoplasmic calcium, promotion of myogenic, or differentiation of basal myogenesis [69]. It is worth emphasizing that Surma et al. (2014) also demonstrated qualitatively similar results to Yap’s results, although different MF was applied (weak static magnetic fields) [70,71]. The example of L929 cell cultures exposed to RMF with frequency *f* = 30 Hz demonstrated the rule that the highest intracellular Ca^2+^ ion concentration had the highest cell migration ability and the wound closure process. The same rule can be found for HaCaT cells (in RMF *f* = 30 Hz and *B* = 13.6 and 18.6 mT). As was noticed, among others, by Hao, the natural process of wound healing involves scaffold/wound implantation. That is why cell migration is crucial for that process; thus, due to cells’ (e.g., fibroblasts) stimulation to grow and migration by mechanical forces, magnetic stimuli are analyzed for further application in therapies for wound healing [72,73,74]. For example, superparamagnetic scaffold and magnetic field were found to promote migration and the closure capacity (not only) of fibroblast but also enhance endothelial cells in angiogenesis [71].

## 4. Materials and Methods

### 4.1. Experimental Set-Up

Figure 9 shows the sketch of the experimental set-up used in the presented study.

The experimental set-up was constructed using a three-phase stator of an induction squirrel cage motor. The RMF source was the stator windings powered by a balanced 3-phase AC power supply. This kind of MF has a constant intensity over time, while it changes its direction continuously at any point in the domain. The cell culture plates (1) were placed in the generator of RMF (2). The proper functioning of the stator with the eliminated rotor required the reduction of the phase-to-phase voltage from 400 V up to 100 V. The frequency of the electrical current *f* (in the case of this work *f* was equal to 30 and 50 Hz) was adjusted using the AC transistorized inverter (3).

The cylindrical glass container (4) was placed coaxially inside the RMF generator. This generator was also placed in the cooling jacket (5) made of stainless steel. The housing of the RMF generator did not have magnetic properties. The space between the inner wall of the jacket (5) and the outer wall of the container (3) was filled with silicone oil, which was used as a medium to remove excess heat produced during the RMF generator operation. Additionally, the circulation pump (6), a heat exchanger (7), and a three-way valve (8) with the automatic temperature controller were used to remove the accumulated heat in the experimental apparatus. The microprocessor temperature sensor (9) and the multifunctional meter (10) were used to control the temperature inside the RMF generator. In the case of the applied experimental set-up, the personal computer (11) was applied to the acquisition and archiving of the experimental data.

The application of alternating current caused the generation of RMF inside the cylindrical container. The basic value enabling the characteristics of the applied field is the spatial distribution of the magnetic induction. The FW Bell 5180 Magnetometer Gauss meter (Magnetic Science Inc, MA, USA) connected with the standard transverse probe (STD18-0404) was used to measure the magnetic field values at different points inside the cylindrical glass container. The magnetic field measurements were carried out as follows: (i) Firstly, the calibration of the Hall probe was conducted using the zero flux chamber (*B* = 0 mT); (ii) The Hall probe was mounted in the RMF generator; (iii) The sketch of measurement points (marked as ○) inside the RMF generator was presented in Figure 10; (iv) The value of the magnetic induction at the selected point inside the RMF generator was detected and recorded. The obtained values of the magnetic field may be presented in the system of the polar coordinate system as the iso-contours patterns. The examples of magnetic induction patterns in the middle part of the experimental set-up are presented in Figure 11. It should be noticed that the maximum values of magnetic induction are in the middle of the RMF generator. These values are decreased toward the center of the RMF generator; cultures of the L929 and HaCaT cell lines were exposed to RMF at a different frequency seeded into 24-well plates, and the plates were placed in the glass container in the middle of the coil. The values of the magnetic induction (*B*) in all wells in the 24-well culture plates were detected by using the FW Bell 5180 Magnetometer Gauss with the probe. Figure 12 shows the obtained values of magnetic induction for the frequency of 30 and 50 Hz. Both cell lines cultured in 24-well plates were placed in the glass container in the center of the coil for 4 h. The sham-exposure samples were incubated for 4 h in the rotating magnetic field generator without RMF activation [3].

It should be noticed that the measured values of magnetic induction in wells of 24-well culture plate have similar numerical values. As can be seen from Figure 13, similar values of magnetic induction are marked with the same color. Based on these measurements, it was decided to calculate the averaged values of magnetic induction. The scheme for calculating this value is given in Table 2. 

The averaged values of magnetic induction for the tested frequency of RMF are collected in Table 1. As follows from this calculation, the obtained averaged values (AV-1, AV-2, AV-3, AV-4, and AV-5) for the tested RMF frequencies are similar. Therefore, the averaged values of magnetic induction for the tested frequencies were obtained (the last column in Table 2). It should be emphasized that these values will be used later in this paper. 

### 4.2. Cell Lines, Cell Culture Conditions, and Experimental Treatment

The effect of the rotating magnetic field (RMF) on cells response was evaluated using in vitro cell line models—L929 (mouse fibroblast cells; ATCC^®^ no. CCL-1^TM^) and HaCaT (human keratinocytes cells; CLS Cell Lines Service no. 300493) [75].

Both cell lines were seeded into 24-well plates (Corning Inc., Corning, NY, USA) at the density of 9 × 10^4^ per well (for CCK-8, NRU, ROS, intracellular concentration of ionized calcium assays) were grown in complete DMEM (Dulbecco’s Modified Eagle Medium, High Glucose) culture medium (Corning Inc., Corning, NY, USA) supplemented with 10% heat-inactivated foetal bovine serum (Corning Inc., Corning, NY, USA), 2 mM L-glutamine (Corning Inc., Corning, NY, USA), 50 IUmL^−1^ penicillin and 50 µgmL^−1^ streptomycin (Corning Inc., Corning, NY, USA) and 10 mM HEPES (Sigma-Aldrich, St. Louis, MO, USA) as was described elsewhere [3,57]. All cell cultures were maintained in standard cell culture conditions at temperature—37 °C, 5% of CO_2_ concentration and relative humidity (RH) at the level of 95%. Cell cultures were monitored with a Nikon TS-100 microscope (NIS Elements F Package version 4.00.06, camera Nikon DS-Fi1, Nikon, Melville, NY, USA) [3,76].

In the presented experiment, modulation of cellular response to different parameters of the rotating magnetic field (RMF) was performed using the self-designed equipment described in detail above (Figure 9 and Figure 10). Cultures of the L929 and HaCaT cells were exposed to RMF at six different magnetic induction values: 5.9, 13.6, 18.6, 22.8 and 28.4 mT (magnetic induction values were presented as a mean for two different frequencies: 30 and 50 Hz). Both cell lines cultured in 24-well plates were placed in the glass container in the center of the coil for 4 h. The sham-exposure samples were incubated for 4 h in the rotating magnetic field generator without RMF activation [3].

### 4.3. Relative Cell Viability

The CCK-8 (Cell Counting Kit-8; Sigma-Aldrich, St. Louis, MO, USA) and neutral red uptake (NRU) assays (In Vitro Toxicology Assay Kit; Sigma-Aldrich, St. Louis, MO, USA) has been used to obtain the cell viability of L929 and HaCaT cell lines analyses after a 4-h exposition to the RMF. The Cell Counting Kit-8 proceeds via the conversion of tetrazolium salt into water-insoluble colored formazan by living cells; thus, the number of living active cells is proportional to the amount of the formazan. Prior to the absorbance detection, the CCK-8 solution (1:10 *v/v*) was added to each well and incubated for 3 h at 37 °C. According to the vendor’s protocol using Synergy H1 Hybrid Multi-Mode Microplate Reader (BioTekWinooski, VT, USA), the absorbance at 450 nm (reference wavelength at 650 nm) was measured. All the experiments were conducted in triplicate. The effect of different parameters of the rotating magnetic field on cellular metabolic activity was calculated using the following Formula (1) as was described elsewhere [3,76]:(1)$$Relative\;viability\;from\;CCK-8\;assay\;\left(\%\right)=\left(\frac{sample\;A_{450-650\;nm}}{positive\;control\;A_{450-650\;nm}}\right)100$$
where *A* is the absorbance.

The cell viability of L929 and HaCaT was also determined using the neutral red uptake assay that provides quantitative information about the number of viable cells that can store neutral red dye in acidic organelles (e.g., in lysosomes) [68]. Complete DMEM medium containing neutral red (1:10 *v/v*) was added to L929 and HaCaT cultures and incubated for 3 h in standard culture conditions. After the incubation, the cells were washed two times with DPBS (Corning Inc., Corning, NY, USA); then, Neutral Red Assay Solubilisation Solution (Sigma-Aldrich, St. Louis, MO, USA) was added to the cells. After 10 min of incubation at room temperature, plates with cells were gently stirred, and the absorbance at 540 nm was measured (the background absorbance of multiwell plates at 690 nm) using Synergy H1 Hybrid Multi-Mode Microplate Reader (BioTek, Winooski, VT, USA)**.** All the experiments were conducted in triplicate. The cell viability was calculated using Equation (2) as was described elsewhere [3,76]:(2)$$Neutral\;red\;uptake\;assay\;\left(\%\right)=\left(\frac{sample\;A_{540-690\;nm}}{positive\;control\;A_{540-690\;nm}}\right)100$$
where *A* is absorbance [3,77].

All the experiments were conducted in triplicate, and in each experiment, culture samples were repeated four times for every magnetic induction (*B*) value.

### 4.4. ROS Measurement

Prior to the level of intracellular reactive oxygen species (ROS) in the L929 and HaCaT cultures measurement, both cell lines were seeded into 24-well black plates (Eppendorf AG, Hamburg, Germany) at the density of 9 × 10^4^ per well and cultured for 24 h. The ROS level was analyzed immediately after a 4-h exposition to RMF using H2DCFDA–fluorogenic dye that measures hydroxyl, peroxyl, and other reactive oxygen species (ROS) activity within the cell (Cellular Reactive Oxygen Species Detection Assay, Abcam, Cambridge, UK). Cell cultures were pre-incubated with 25 µM DCFDA for 45 min at 37 °C in dark, and the fluorescence was analyzed immediately with a microplate reader (at Ex/Em = 485/535 nm) Synergy H1 Hybrid Multi-Mode Microplate Reader (BioTek, Winooski, VT, USA). All the experiments were conducted in triplicate.

### 4.5. Intracellular Concentration of Ca^2+^ Assay

The modulation of L929 and HaCaT in response to RMF was also determined by the intracellular concentration of ionized calcium measurement using the colorimetric Calcium Detection Kit (Abcam, Cambridge, UK). After a 4-h exposition to RMF, cells cultured in 24-well plates (Corning Inc., Corning, NY, USA) were harvested and prepared for assay procedure, as described by the manufacturer’s protocol. Detection of the absorbance was performed at 575 nm using Synergy H1 Hybrid Multi-Mode Microplate Reader (BioTek, Winooski, VT, USA). Calcium concentrations were determined based on the standard curve using the following Equations (3) and (4):(3)$$Sa=\left(\frac{Corrected\;absorbance-y\;intercept}{Slope}\right)$$
(4)$$Calcium\;concentration=\left(\frac{Sa}{Sv}\right)D$$
where *Sa* is a sample amount (in μg) from standard curve; *Sv*—sample volume (μL) added into the wells; *D*—dilution factor.

All the experiments were conducted in triplicate.

### 4.6. In Vitro Wound Healing Assay

Moreover, wound healing assays were performed to determine the modulatory cellular response to RMF on wound closure efficiency. L929 and HaCaT were grown in 24-well plates at a density of 1 × 10^6^ per well and were incubated for 24-h. Then, linear scratches were made in the confluent L929 and HaCaT monolayer cultures with a sterile p100 pipet tip [78,79]. Then, cells were immediately rinsed with DPBS to remove cellular debris, and images of the scratches were taken (at t = 0 h) using the digital camera (NIS Elements F Package version 4.00.06, camera Nikon DS-Fi1, Nikon, Melville, NY, USA) connected to the inverted microscope Nikon TS-100 (Nikon, Melville, NY, USA). Next, cell cultures were exposed to RMF for 4 h. The images of the scratches were taken 24 and 48 h later and were analyzed with ImageJ software version 1.52r (National Institutes of Health, Bethesda, MD, USA). Experiments were conducted in triplicate, and in each experiment, culture samples were repeated four times for every magnetic induction (*B*) value.

### 4.7. RNA Isolation, Reverse Transcription Reaction, and Real-Time PCR Analyses

Finally, the expression of genes involved in processes crucial in wound healing (e.g., cell migrations and ECM synthesis, Table 3) in both analyzed cell lines after cell exposition to RMF was determined using the real-time PCR method (real-time PCR experiments were conducted according to the MIQE guidelines [80]). Total RNA from L929 and HaCaT cell cultures was isolated immediately after 4-h RMF exposition. For total RNA isolation, the L929 and HaCaT cell cultures were washed three times with DPBS (Corning Inc., Corning, NY, USA) and removed from the culture surface using tissue cell scrapers (TPP Techno Plastic Products AG, Trasadingen, Switzerland). The cell pellets were obtained by centrifugation (1000 rpm; 5 min; 24 °C), preserved in RNAlater (Sigma-Aldrich, St. Louis, MO, USA), and storage at −80 °C until total RNA isolation. 

Total RNA was extracted using InviTrap^®^ Spin Cell RNA Mini Kit (Invitek Molecular GmbH, Berlin, Germany) following the manufacturer’s protocol. Prior to reverse transcription, RNA quality was determined via horizontal electrophoresis through 1.5% agarose gel (MicroporOmega Agarose, Prona Agarose, EU), whereas the total RNA concentration of was determined using the Quant-iT^TM^ RNA HS Assay Kit and a Qubit fluorometer (Thermo Fisher Scientific, Waltham, MA, USA). Prior to RT-PCR, genomic DNA was removed from RNA isolates by DNase I digestion (1U/1 μg of RNA; Thermo Fisher Scientific, Waltham, MA, USA). 

The reverse polymerase chain reactions (RT-PCR) were performed using total RNA (40 ng) and the RevertAid™ Premium First Strand cDNA Synthesis Kit (Thermo Fisher Scientific, Waltham, MA, USA) according to the manufacturer’s instruction. Each RNA sample was reverse-transcribed in triplicate. The single-stranded cDNA samples were preserved at −80 °C until real-time PCR reactions. Prior to, qPCRs single-strand cDNA samples pooled for qPCR analyses and quantified using Quant-iT^TM^ ssDNA Assay Kit and Qubit fluorometer (Thermo Fisher Scientific, Waltham, MA, USA). The qPCRs were performed in a Rotor-Gene Q instrument (Qiagen GmbH, Hilden, Germany) using mixtures containing cDNA (pooled 100 ng), 400 nM of each primer (primers sequences were designed to span an exon-exon junction using Primer3 Input software version 4.1.0; https://primer3.ut.ee; accessed on 12 July 2018)—Table 3 and synthesized by Genomed S.A., Warsaw, Poland), FastGene^®^IC Green qPCR Universal Mix (Kapa Biosystems, Woburn, MA, Germany), and nuclease-free water (Thermo Fisher Scientific, Waltham, MA, USA) in a total volume of 15 µL (all samples were run in triplicates; no template control (NTC) was included in each run). Real-time PCR conditions included initial heating at 95 °C for 2 min, followed by 40 cycles of amplification (denaturation at 95 °C for 5 s, annealing and elongation at 60 °C for 30 s). Primer sequences for reference and experimental genes were confirmed to be highly specific (single peaks in melting curve analyses were observed; Appendix A) by melt curve analysis of PCR amplicons (Rotor-Gene Q Series Software 2.3.1, Qiagen GmbH, Hilden, Germany). 

The relative expression of the analyzed genes was calculated according to the 2^−ΔΔCt^ method [81,82], where glyceraldehyde 3-phosphate dehydrogenasegene (GAPDH) was used as reference genes.

### 4.8. Statistical Analyses

The data obtained in this study were presented as mean values ± standard deviation (SD) and statistical analyses were performed using the one-way analysis of variance—ANOVA (STATISTICA 13.3 statistical software, StatSoft Inc., Tulsa, OK, USA). The post hoc Tukey’s HSD (honest significant differences) method was used in ANOVA for data obtained from all performed assays as was described elsewhere [83]. Differences were considered significant *p* < 0.05 for all assay results obtained in this study (Appendix A). The multiple correlation coefficient (*R*) between the level of ROS, Ca^2+^, and the rate of wound healing was carried out as was described elsewere [84] using Statistica 13.3 software.

## 5. Conclusions

Our study showed that different values of *B* and *f* of RMF evoke the different response of cells. The observed in our study increase in general metabolic activity (mainly for HaCaT cells and partially for L929 cells) may be associated with the increase of ROS levels. The human keratinocytes exposed to RMF (*f* = 50 Hz and *B* = 13.6 and 28.4 mT) exhibited the lowest ability in wound healing, and those cultures displayed lower ROS levels (but higher than in control samples) and lower intracellular Ca^2+^ ion concentration. Interestingly, lower intracellular Ca^2+^ concentration (for RMF at the frequency of 50 Hz) is reflected in the inability of cells to wound closure. In the case of L929 cells, the highest rate in the wound healing process had the lowest magnetic induction values. Thus, it can be stated that the subtle balance in the ROS level is crucial in the wound/injured tissue for effective healing process and regeneration, and is it possible to modulate the cellular response to different parameters of the RMF in the context of an in vitro wound healing. The relative expression level of *col1a1* and *col3a1* genes differ in both groups of cells exposed to 30 and 50 Hz. The highest expression level (RQ) was described for the *RAC1* gene, which is involved in the migration process. Based on performed analyses and obtained results, it may be concluded that the RMF for frequency 30 Hz (magnetic induction equal to 13.6 and 18.6 mT) gives generally better results at different levels of cellular activities; that is why further studies should be conducted to evaluate the mechanism of RMF’s action on the cell in vitro models.

## Figures and Tables

**Figure 1 ijms-22-05785-f001:**
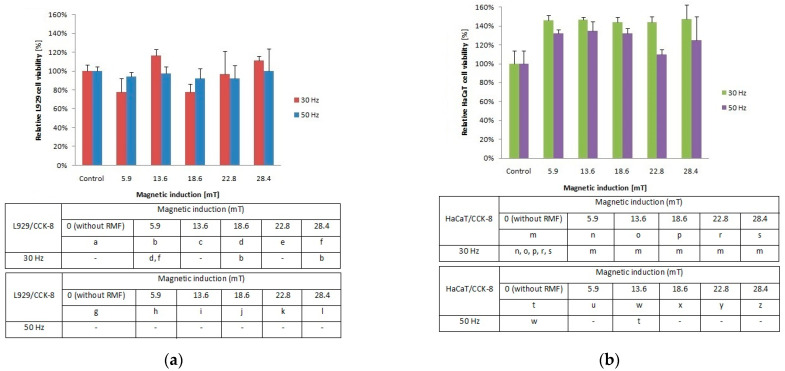
Relative viability after 4-h exposition to different magnetic induction and frequency of RMF. (**a**) CCK-8 assay results for L929 cell line; (**b**) CCK-8 assay results for HaCaT cell lines (all samples were compared to control; *p*-values < 0.05 were considered significant and were represented by small letters).

**Figure 2 ijms-22-05785-f002:**
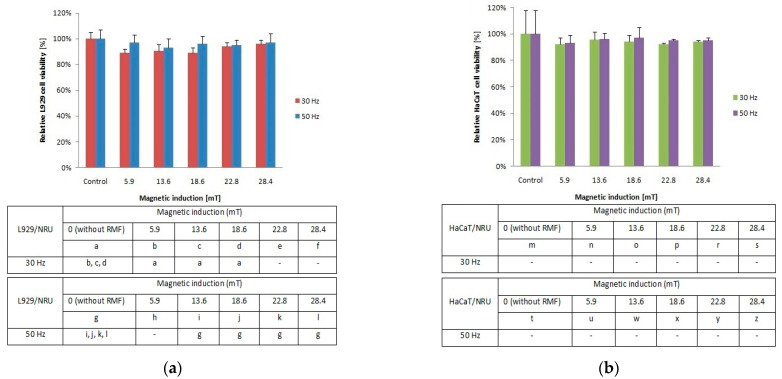
Relative viability after 4-h exposition to different magnetic induction and frequency of RMF. (**a**) NRU assay results for L929 cell line; (**b**) NRU assay results for HaCaT cell line lines (all samples were compared to control; *p*-values < 0.05 were considered significant and were represented by small letters).

**Figure 3 ijms-22-05785-f003:**
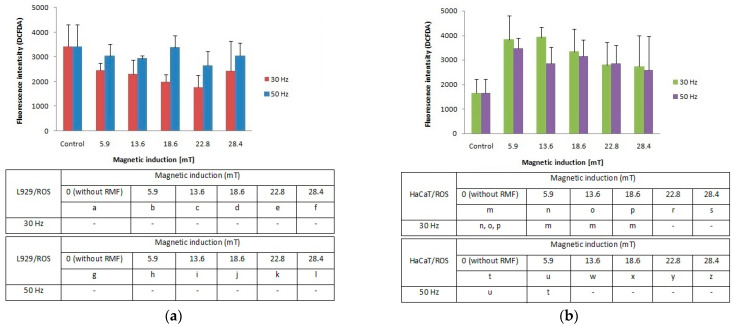
ROS accumulation (fluorescence intensity; DCFDA) in L929 (**a**) and HaCaT (**b**) cell cultures after 4-h exposition to different magnetic induction and frequency of RMF (all samples were compared to control; *p*-values < 0.05 were considered significant and were represented by small letters).

**Figure 4 ijms-22-05785-f004:**
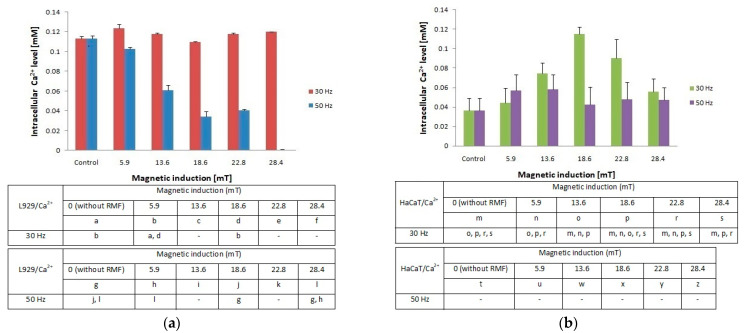
Intracellular Ca^2+^ level in L929 (**a**) and HaCaT (**b**) cell cultures after 4-h exposition to different magnetic induction and frequency of RMF (all samples were compared to control; *p*-values < 0.05 were considered significant and were represented by small letters).

**Figure 5 ijms-22-05785-f005:**
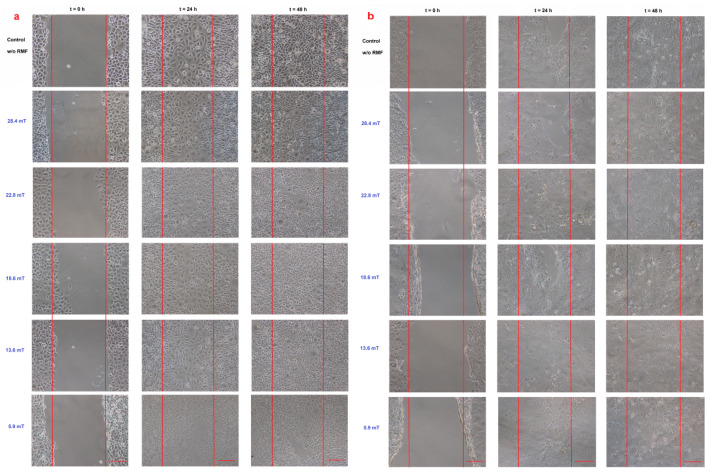
Wound healing assay (**a**) L929 cell line and (**b**) HaCaT cell line in different magnetic induction values; scale bar = 50 μm.

**Figure 6 ijms-22-05785-f006:**
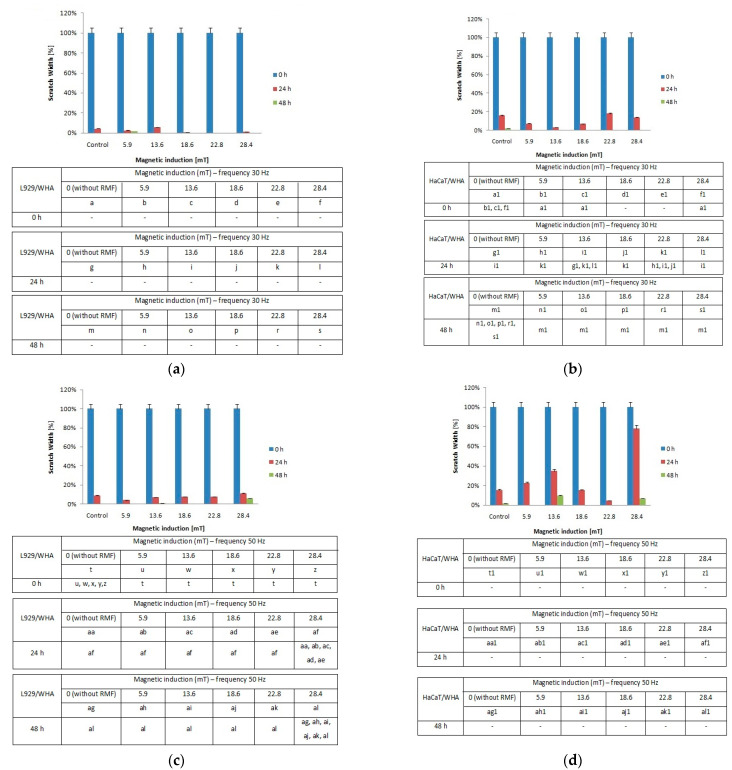
The wound closure percentage in wound scratch assay after 24 and 48 h from 4-h exposition to RMF (measured by ImageJ software). (**a**) L929 cells—frequency 30Hz; (**b**) HaCaT cells—frequency 30 Hz; (**c**) L929 cells—frequency 50 Hz; (**d**) HaCaT cells—frequency 50 Hz (all samples were compared to control; *p*-values < 0.05 were considered significant and were represented by small letters).

**Figure 7 ijms-22-05785-f007:**
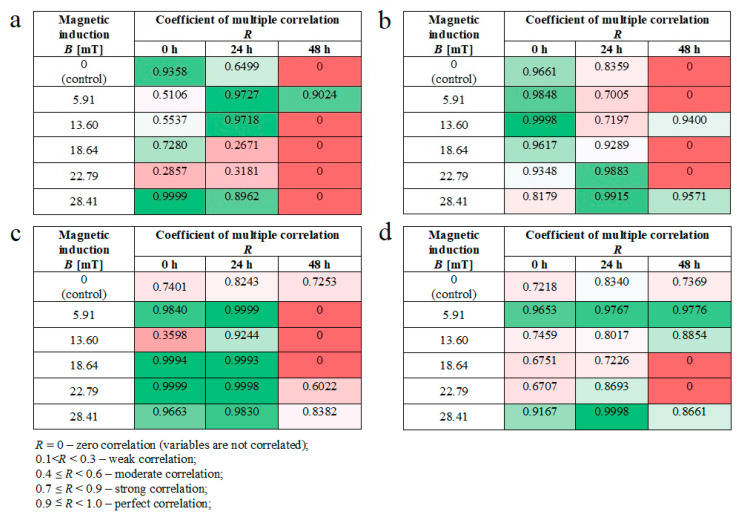
The values of correlation coefficients for L929 and RMF at frequency equal to 30 Hz (**a**); for L929 and RMF at frequency equal to 50 Hz (**b**); for HaCaT and RMF at frequency equal to 30 Hz (**c**); for HaCaT and RMF at frequency equal to 50 Hz (**d**). Differneces in correlation represented by different colors: zero correlation–dark red; weak correlation–red; moderate correlation–light red; strong correlation–light green; perfect correlation–dark green.

**Figure 8 ijms-22-05785-f008:**
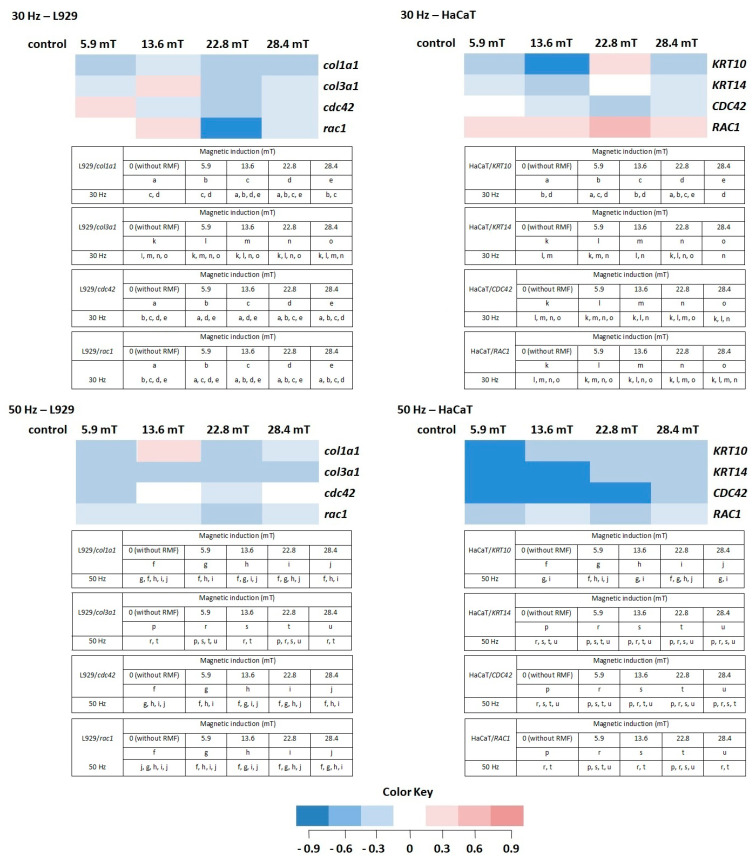
The profile of gene (involved in the wound healing process) expression in L929 and HaCaT cell cultures after exposure to RMF (all samples were compared to control; *p*-values < 0.05 were considered significant and were represented by small letters). Differneces in gene expression level represented by different colors: expression level equal to control—white; lower expression level—light blue; medium lower expression level—blue; the lowest expression level—drak blue; higher expression level—light red; medium higher expression level—red; the highest expression level—dark red.

**Figure 9 ijms-22-05785-f009:**
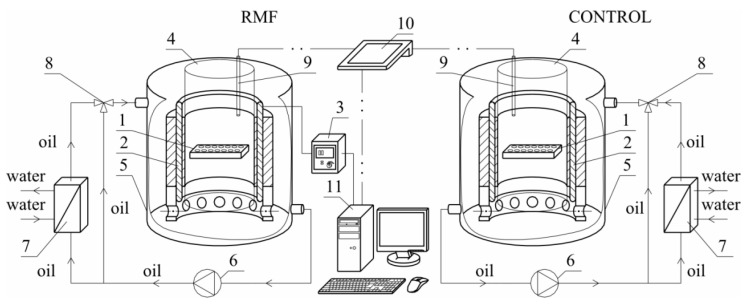
The sketch of experimental set-up: 1—cell culture microplate, 2—RMF generator, 3—AC transistorized inverter, 4—cylindrical glass vessel, 5—cooling jacket, 6—circulating pumps, 7—heat exchanger, 8—three-way valve, 9—microprocessor temperature sensor, 10—multifunctional meter, 11—personal computer.

**Figure 10 ijms-22-05785-f010:**
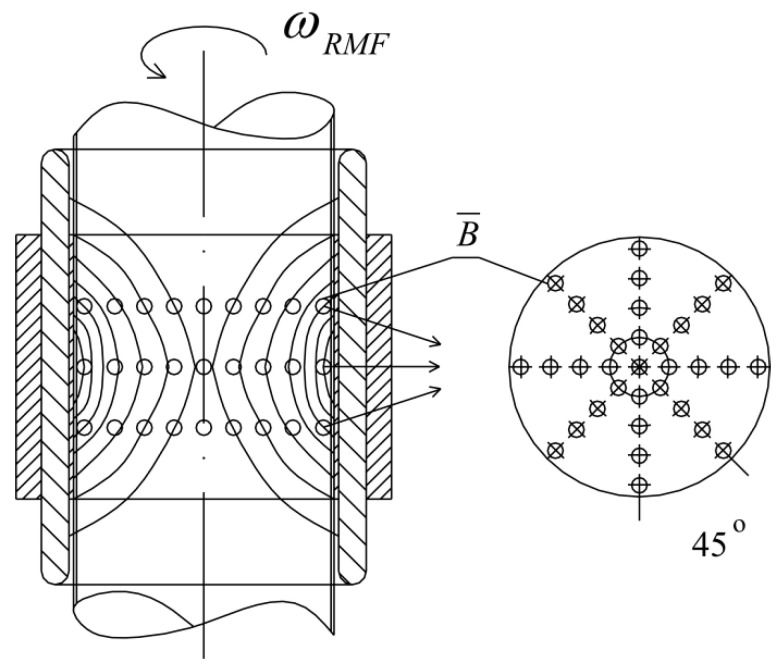
The arrangement of measurement points of magnetic induction inside the RMF generator (these points are compatible with the points where the probe was placed).

**Figure 11 ijms-22-05785-f011:**
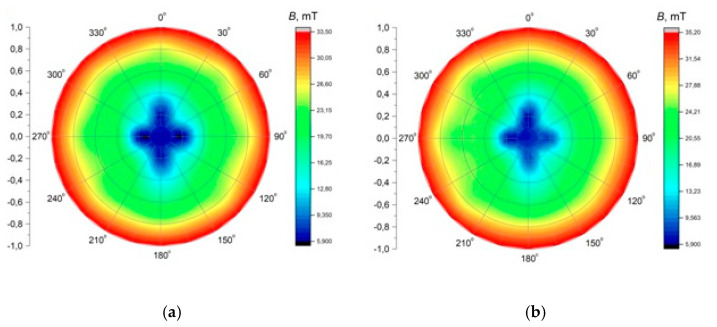
The contour patterns of the spatial distribution of the magnetic field in the selected cross-section of the RMF generator (in the middle of set-up) for *f* = 30 Hz (**a**) and *f* = 50 Hz (**b**).

**Figure 12 ijms-22-05785-f012:**
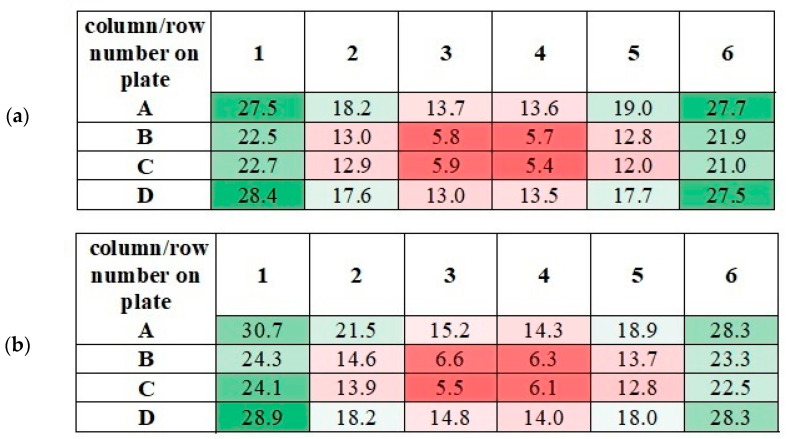
The measured values of the magnetic induction in the wells of 24-well culture plates for *f* = 30 Hz (**a**) and *f* = 50 Hz (**b**). All values are given in militesla (mT). Differneces in magnetic induction repersented by different colors: the weakest magnetic iduction–dark red; weak magnetic induction–light red; moderate magnetic indcution–light green; the highest magnetic induction–dark green.

**Figure 13 ijms-22-05785-f013:**
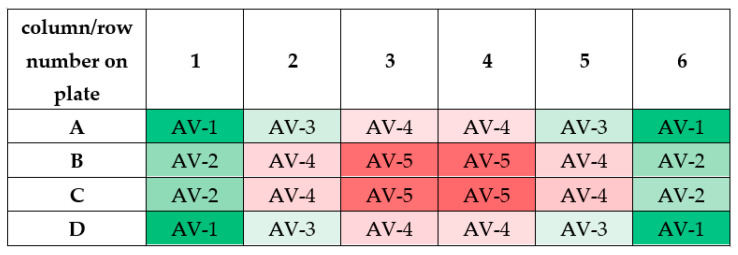
The scheme for calculating the averaged values of magnetic induction (abbreviation AV means ”averaged value”; AV-1 means that the averaged value of magnetic induction was calculated based on the measurements taken in these wells). Differneces in magnetic induction repersented by different colors: the weakest magnetic iduction–dark red; weak magnetic induction–light red; moderate magnetic indcution–light green; the highest magnetic induction–dark green.

**Table 1 ijms-22-05785-t001:** Characteristics of MFs.

Constant Magnetic Field (CMF)		Alternating Magnetic Field (AMF)			
Static MF (SMF)	Magnetostatic field (MSF)	Sinusoidally changing MF	Pulsed magnetic field (PMF)	Pulsating magnetic field (PuMF)	Geomagnetic field (GMF)
Magnetization of a permanent magnet or electromagnet coils supplied with unidirectional current		Electromagnetic coils supplied with a current of 50 Hz or other frequencies	Electromagnetic coils supplied with current pulses	Electromagnetic coils supplied with current from a rectifier data	MF of the Earth plus the MF of the ionosphere

**Table 2 ijms-22-05785-t002:** The averaged values of magnetic induction for the tested frequency of RMF.

No.	*f* = 30 Hz	*f* = 50 Hz	Averaged Values of Magnetic Induction for Tested Frequencies
AV-1	27.8	29.1	28.4
AV-2	22.0	23.6	22.8
AV-3	18.1	19.2	18.6
AV-4	13.1	14.2	13.6
AV-5	5.7	6.1	5.9

**Table 3 ijms-22-05785-t003:** Set of primer sequences used for real-time PCR analysis.

Genes	Primer Sequences (5’ → 3’) ^1^	Amplicon Size (bp)	Accession No.
**Reference genes**			
**L929 cells**			
*actb*	F-GGCACCACACCTTCTACAATG R-GGGGTGTTGAAGGTCTCAAAC	133	XM_030254057.1
*gapdh*	F-AACTTTGGCATTGTGGAAGG R-GGATGCAGGGATGATGTTCT	132	XM_017321385.2
**HaCaT cells**			
*ACTB*	F-CTCTTCCAGCCTTCCTTCCT R-AGCACTGTGTTGGCGTACAG	116	NM_001101.5
*GAPDH*	F-AAGGTGAAGGTCGGAGTCAA		
	R-AATGAAGGGGTCATTGATGG	108	NM_002046.7
**Genes of interests (GOIs)**			
**L929 cells**			
*col1a1*	F-CGATGGATTCCCGTTCGAGT R-GAGGCCTCGGTGGACATTAG	96	NM_007742.4
*col3a1*	F-GACCTAAGGGCGAAGATGGC R-GAAGCCACTAGGACCCCTTTC	95	NM_009930.2
*cdc42*	F-CCCTCACACAGAAAGGCCTAAA R-ATGCGTTCATAGCAGCACAC	107	XM_030253140.1
*Rac1*	F-GGAGACGGAGCTGTTGGTAA R-TTGTCCAGCTGTGTCCCATA	156	NM_001347530.1
**HaCaT cells**			
*KTR 10*	F-TGATGTGAATGTGGAAATGAATGC R-GTAGTCAGTTCCTTGCTCTTTTCA	147	NM_000421.5
*KTR 14*	F-GGCCTGCTGAGATCAAAGACTAC R-CACTGTGGCTGTGAGAATCTTGTT	80	NM_000526.5
*CDC42*	F-AGGCTGTCAAGTATGTGGAG R-TCATAGCAGCACACACCTGC	128	NM_001039802.2
*RAC1*	F-CCCTATCCTATCCGCAAACA R-CGCACCTCAGGATACCACTT	100	NM_006908.5

^1^ For qPCR all primers sequences were designed to span an exon-exon junction using Primer3 software.

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
