# Peer review of "Modulation of Cellular Response to Different Parameters of the Rotating Magnetic Field (RMF)—An In Vitro Wound Healing Study"

_ijms, 2021, doi:10.3390/ijms22115785_

Round 1
Reviewer 1 Report
The manuscript explores the effect of different values of magnetic induction and frequencies of rotating magnetic field on fibroblasts and keratinocytes. The study of the potential use of electromagnetic fields for wound healing is very interesting. Several in vitro techniques were performed on both cell types, including cell viability, ROS generation, intracellular calcium, migration and gene expression of specific markers. The study is well designed and performed, obtaining interesting information about the effect of magnetic fields on cells involved in wound healing. However, the data processing and statistical analysis showed several issues that should be addressed.
- The statistical analysis performed was ANOVA with the post hoc Fisher’s LSD. Why did the authors use the Fisher’s LSD test and not another like Bonferroni or Tukey?
- For each assay authors used a different significant p value (p<0.2, p<0.15 or p<0.05). It is commonly used p<0.05 for the significance of in vitro analyses. Why did the authors used p<0.2 for CCK8 measurements and p<0.15 for ROS data? The authors should specify the reason at the manuscript.
- Supplementary figures include the statistical differences among conditions. However, the comparisons are not clearly described. Figure captions explain “all samples were compared to control; p-values˂0.05 were considered significant and were represented by small letters”, although it seems that also comparisons between conditions are shown. In addition, the same letters are used for 30 Hz and 50 Hz, although, according to the data, the comparisons are done separately.
- The authors used two cell types from different origin. Could the interspecies differences influence the behaviour observed?
- The cell viability was analysed using two methods. CCK-8 assay showed significant differences whereas NRU assay did not show differences. Both assays are considered cell viability assays, so was the viability reduced or proliferation enhanced by the magnetic induction? As the authors discussed, rotating magnetic fields could be responsible of enzymatic activity changes of cells, although the experiment performed is not specific for enzymatic activity. According to the results obtained, it is not possible to elucidate if the enzymatic activity was modified or the cell viability was affected. Cell number quantification or specific enzyme activity measurements could clarify the point.
- The gene expression of almost all markers was reduced due to magnetic field. The authors should discuss the potential effect of magnetic field in gene expression.
- The experiments performed allowed to obtain a high amount of results, in some cases difficult to understand the overall effect. A discussion summarizing the main results and describing the potential implications of the effect of rotating magnetic fields on both cell types analysed is necessary.
- Some acronyms like MF are defined twice and others like WMF are not defined. Please check them carefully.
- In line 204, col3a1 should be col1a1.
Reviewer 2 Report
Dear Editor!
The article is devoted to an urgent problem - the study of the influence of different parameters of the magnetic field on the processes of wound healing. The relevance is associated with the successful use of a magnetic field for the treatment of human wounds. The cellular viability, ROS and Ca2 + concentration levels, wound healing assay and gene expression analyzes were conducted to evaluate effect of RMF. Notes: 1) Data on the role of ROS and Ca2 + in the regulation of the functional state of cells are well known. The authors conclude that the changes in the level of ROS and Ca2 + in cells under the influence of a magnetic field are correlated with the rate of wound healing, but they do not provide any calculations. To substantiate the conclusion, it is necessary to calculate the correlation coefficients. 2) Insufficiently substantiated the choice of proteins, the expression of which was studied. Based on these data, it is impossible to answer the question about the causes of wound healing. Is it associated with an increase in proliferative activity or cell migration ?. 3) Insufficiently well and fully presented results. 4) There is no scientific hypothesis about the mechanisms of the influence of the magnetic field on the wound healing processes. Suggestions: 1) In the introduction, present a scientific hypothesis about the mechanisms of the influence of the magnetic field on the wound healing processes. 2) Calculate the correlation coefficients between the level of ROS, Ca2 + and the rate of wound healing. 3) Transfer photographs of cell cultures from Supplementary Materials to the main text. 4) Based on the results obtained, present the proposed mechanism of the influence of the magnetic field on the wound healing process.
Round 2
Reviewer 1 Report
The authors have replied almost all my comments satisfactorily. However, an important point should be clarified before considering the manuscript for publication. The cell viability experiment was performed using CCK-8 assay. According to the experimental section, cells grown on culture plates and exposed to rotating magnetic field generator for 4 hours. Then, immediately after exposure, the CKK-8 assay was performed, with extra 3 hours of incubation. The relative viability is considered as a percentage of live cells compared to a control without treatment. In that case, how could the viability be higher after a treatment? According to the method, this could happen if the cells are able to proliferate faster due to the treatment. However, as the authors discussed on previous response, due to the short time it is not possible to observe changes in proliferation. In this regard, the results analysis is complex and the only possible explanation is that rotating magnetic fields enhance dehydrogenase activity of cells. I recommend rewriting results and part of the discussion to clarify this point. The method is not measuring enzymatic activity, and also the results are not about cell viability. My opinion is that cell viability is not deduced due to the rotating magnetic field and the higher values observed are related to the potential increase enzymatic activity, although the method use is not specific for enzymatic activity. In addition, I recommend including the tables created for statistical analyses into the main manuscript. It is difficult to follow the results without the statistics. Finally, equations have low quality in the recent version. Please check it. The decision is major revision.
Reviewer 2 Report
Dear Editor! The authors took into account all the comments and made the appropriate edits. The manuscript can be published in present form.
Round 3
Reviewer 1 Report
The authors have replied all my comments. I consider the manuscript for publications without any further revision.